# Transit Travel Community Detection and Evolutionary Analysis: A Case Study of Shenzhen

Jingjing Yan [1,2,3], Zhengdong Huang [1,2,3,*], Tianhong Zhao [1,2,3] , Ying Zhang [1,2,3] and Fei Chang [1,2,3]

1    Research Institute for Smart Cities, School of Architecture and Urban Planning, Shenzhen University, Shenzhen 518060, China; yanjingjing2020@email.szu.edu.cn (J.Y.); zhaotianhong2016@email.szu.edu.cn (T.Z.); y.zhang@szu.edu.cn (Y.Z.); changfei2020@email.szu.edu.cn (F.C.)
2    Guangdong-Hong Kong-Macau Joint Laboratory for Smart Cities, Shenzhen 518060, China
3    Shenzhen Key Laboratory of Urban Digital Twin Technology, Shenzhen 518060, China
*    Correspondence: zdhuang@szu.edu.cn

**Abstract:** Community detection can reveal specific urban spatial structures related to human activities, and is achieved using mobility data from various sources. In the existing research, less attention has been devoted to communities related to urban transit travel. As public transit is a key component of the urban transport system, it is important to understand how transit communities are organized and how they evolve. This research proposes an approach to urban transit travel community detection using transit travel data and examines how the communities have evolved over time. The results in Shenzhen from 2015 to 2017 showed that the transit travel network had an obvious community structure, and the components (TAZs in this case) of the communities changed over time. During the three years, the western part of Shenzhen experienced more component changes on weekdays, and the central part of the city underwent more component changes on weekdays. In addition, the transit travel communities had a significant coupling relationship with urban administrative divisions. Exploring transit travel communities provides insight for improving public transit systems and enriches the research genealogy of urban spatial structure.

**Keywords:** travel network of transit; community structure; transit smart card; dynamic evolution; Shenzhen

## 1. Introduction

The spatial structure of a city is the spatial expression of the interaction among various elements such as the material environment, socio-economic factors, and transportation facilities in the city; it has always been the focus of research in the field of urban planning. In recent years, with the advancement of urbanization, the "space–time compression" effect caused by rapid social economy and transit development has significantly enhanced the spatiotemporal convenience of urban travel [1–3]. The functionality and network connectivity of cities has been strengthened, leading to the rapid evolution of urban spatial structures [4–7]. Analyzing urban spatial structures and their evolution from a macro perspective can help improve insights into sustainable urban space development, thus allowing for intelligent spatial decision making in urban planning and management [8,9]. It is also an important scientific path for promoting and achieving inclusive sustainable urbanization.

Traditional urban spatial structure planning is mainly based on natural topography, transportation infrastructure, land use properties, etc. Studies of these elements are combined with the qualitative research and judgment derived from the subjective analyses of planners to mainly identify the "static" state of the physical conditions [10]. The development of information and communications technology (ICT) has enabled the collection of large-scale, high-resolution spatiotemporal big data. Big data have transformed traditional research on community structures [11,12] and promoted the development of service-oriented cities and urban science. Spatial flow theory based on big data has become the core

theory used to guide the reform of urban spatial planning methods [13]. Macro-, meso-, and micro-scale problems in the urban development process can be visually expressed using large volumes of dynamic and accurate individual spatial data, thus providing important support toward the goal of constructing "people-oriented" cities [14] and also providing new methods and ideas for the study of urban spatial structures.

Coupling spatiotemporal big data and complex network-based community detection technology has attracted the attention of many studies in the field of urban spatial structures. Community detection is based on the premise that nodes in the same community are closely linked, whereas nodes in different communities are only loosely linked [15–17]. This method can be applied to different spatial scales. At the national level, Ratti analyzed and verified national administrative boundaries by comparing existing norms with community structures in interaction-based networks [18]. At the regional and urban levels, de Monti used a traditional survey based on OD data to determine the community structure of an island region in Italy and provided recommendations for sub-regional planning [19]. Liu used taxi GPS traces to reveal the hierarchical structure of urban areas in Shanghai, China, and identified a two-level polycentric urban structure [20]. In addition, there are also studies that analyze the dynamic changes in communities at various times of day [21].

Analyzing urban spatial structure from the perspective of public transportation is meaningful for urban sustainable development. On the one hand, urban public transportation is a complex network that plays an important role in building sustainable cities and communities and plays an important role in guiding and supporting urban spatial structure. On the other hand, public transportation can demonstrate the matching effect and guiding role of urban public transportation on residential and employment needs and has reference value for formulating policies regarding coordinated development of urban land use and public transportation. However, existing research mostly uses mobile phone signaling data and taxi GPS data as the main data sources for exploring community structures. There is a lack of research on urban spatial structure from the perspective of public transportation. In addition, from a time-scale perspective, the existing research mainly analyzes urban spatial community structures during a specific period and pays less attention to the dynamic evolutionary characteristics of urban spatial communities over different years.

To fill these gaps in the literature, this paper uses intelligent public transportation card data to analyze the dynamic evolutionary characteristics of urban public transportation communities from a temporal dimension. The term "community" in this article is used to refer to the bus travel community unless otherwise noted. The main research contents of this article include: (i) the passenger flow network of urban transit is constructed for both weekdays and weekends; (ii) based on community detection methods, the community structure of urban transit travel across different years is revealed; (iii) the dynamic evolution characteristics of urban transit travel communities are analyzed in terms of similarity and diversity; and (iv) a comparative analysis of the community detection results based on urban administrative units from 2015 to 2017 is demonstrated, where the term "urban administrative units" refers to the regions divided by the state for the convenience of administrative management.

## 2. Study Area and Data

### 2.1. Study Area

Shenzhen is the central city of the Guangdong–Hong Kong–Macao Greater Bay Area and an important window into China's reform and opening up. It has been established as a comprehensive transport hub with a developed public transport network. Shenzhen city provides a typical geographical transport environment for conducting research on the dynamic spatial structure of urban public transport travel.

Shenzhen is divided into 491 TAZs (Figure 1). In this study, TAZs were selected as the analysis units as they consider the road network and retain similar land usage types and socio-economic attributes at a finer spatial scale than street or administrative district units, which are more suitable for macro-scale research in cities [22]. The base maps of administrative districts

in Shenzhen used in this study were derived from the public version (2021) of the 1:1 million scale basic geographic information data authorized by the Ministry of Natural Resources and provided by the National Geographic Information Resource Directory Service System.

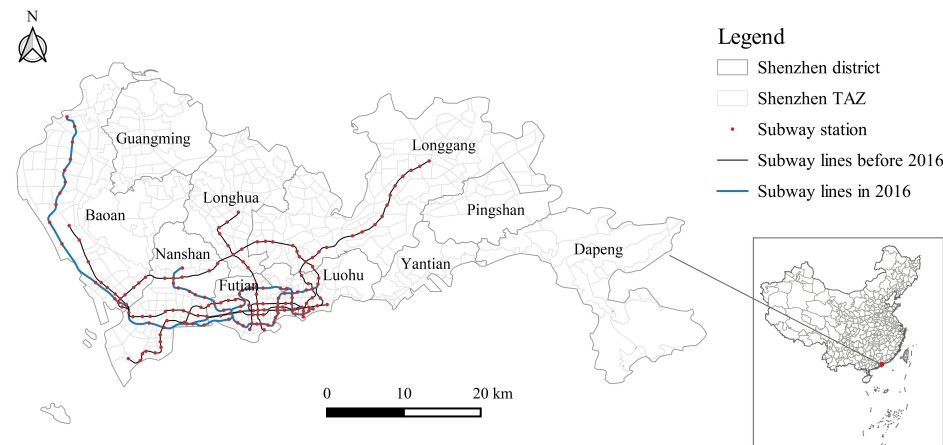

**Figure 1.** Administrative districts and subway lines in Shenzhen.

## 2.2. Data Source and Preprocessing

Residents' travel behaviors within a one-day or one-week period display strong generalization and spatiotemporal laws that can be used to predict daily changes in residents' travel over longer periods [23,24]. With the rapid development of intelligent transportation systems, transit travel demand can be collected through smart card systems [25]. This paper uses smart card data from 2015 to 2017 in Shenzhen for the analysis of the evolution of transit travel communities. Transit travel communities are subject to changes in the city's underlying transit system, such as the opening of new subway lines. By the end of 2015, five rail transit lines in Shenzhen were completed and open to traffic (lines 1–5). By the end of 2016, lines 7, 9, and 11 were also fully completed and open to traffic. Using 2015–2017 as the research period, the dynamic evolution process of urban transit travel communities can be completely observed in the three periods before, during, and after the opening of these lines, which is sufficiently representative.

Therefore, relatively complete records of transit smart card data from 2015 to 2017 (except for the Mid-Autumn Festival, National Day, and other holidays) were collected, and the structural characteristics and evolution of communities were analyzed from the perspectives of both weekdays and weekends. The studied weekdays included 2–6 November 2015, 26–30 September 2016, and 25–29 September 2017. The studied weekends included 28–29 November 2015, 10–11 September 2016, and 16–17 September 2017. All the three-year data used in this study refer to the average daily data based on the above dates.

The transit smart card data from Shenzhen used in this study include metro card data and bus card data. The metro card data include information such as card ID, type of entry and exit, inbound swipe time, outbound swipe time, metro line and metro station information, which can be used to directly obtain the origin–destination (O-D) of passengers' travel. However, the bus card data include only card ID, check-in swipe time, bus license plate number, and bus line name without information regarding passengers' alighting times or locations. Hence, it is essential to deduce the O-D of bus passengers' travel by integrating transit smart card data, bus GPS data, and bus station locations [26,27]. The main steps are as follows:

(1) Data preprocessing: data cleaning was performed by removing abnormal values and deleting duplicate data. The abnormal values mainly include abnormal IC card data values and abnormal GPS values. Among them, the abnormal IC card data values include duplicate data, missing fields, and jumps in swipe time, and the abnormal bus GPS values include abnormal positioning data and skewed GPS data.

(2) Starting station extraction: the arrival timetable of each bus was obtained using GPS data and station data, and the smart card payment time was matched with bus arrival times to identify the boarding stations at which passengers used their cards to pay.

(3) Terminal station deduction: there are four situations, including no transfer and bus-to-bus, bus-to-metro, and metro-to-bus transfers. To resolve the first scenario, historical data were required. For example, for commuters using public transport, the boarding station on the same line for the return journey in the afternoon could be reasonably assumed to be the alighting station for the morning journey. In the second case, the alighting station of the passenger on the first line can be identified; however, the alighting station of the second line or further transfer lines after the transfer is the passenger's true destination. Therefore, the first method must be employed to infer the final alighting station. In the third case, the destination can be readily determined from the metro entry and exit records. In the fourth case, the process described above for the first case is also required to determine the final alighting station of the bus line.

## 3. Methods

### 3.1. Methodology Overview

In this paper, the transit smart card data, bus GPS data, and data from the lines and stations were preprocessed to obtain the O-D matrix of passengers' transit travel (Figure 2). On this basis, the structure and evolution characteristics of urban transit travel communities are analyzed, mainly including: (1) constructing an urban transport travel network model with a TAZ as the analysis unit and analyzing its basic network characteristics, (2) analyzing the structure of urban transit travel communities based on the community detection method, (3) analyzing the community evolution characteristics based on similarity index and visualization analysis, and (4) coupling analysis of transit travel communities with urban administrative divisions based on bubble diagrams.

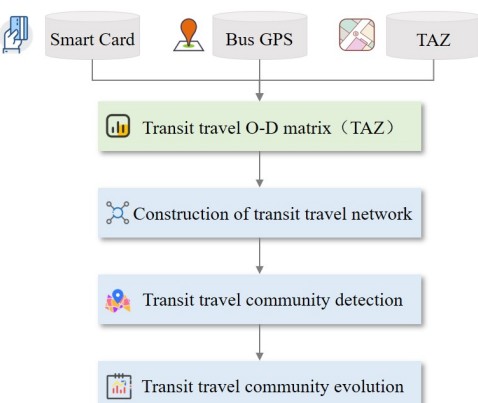

**Figure 2.** Overview of methodology.

### 3.2. Transit Travel Network Construction

The activity tracks of all individuals by transit travel were included in the urban transit travel data set. The transit travel volumes between each TAZ in the city were obtained by aggregating the spatial travel activity in the selected spatial units (i.e., TAZs) and converting the travel between different stations into travel between different TAZs. In this context, $i$ and $j$ are used to denote stations, and $S_{i,j}$ represents the passenger flow between $i$ and $j$. When $i$ and $j$ are spatially located in TAZs $m$ and $n$, respectively, the transit passenger flow between $m$ and $n$, i.e., $T_{m,n}$ can be expressed as:

$$T_{m,n} = \sum S_{i,j} \cdots (i \in m, j \in n) \tag{1}$$

The O-D matrix for urban transit travel based on TAZ can be expressed as:

$$OD_{TAZ} = \begin{bmatrix} T_{1,1} & T_{1,2} & \cdots & T_{1,m} \\ T_{2,1} & T_{2,2} & \cdots & T_{2,m} \\ \vdots & \vdots & \ddots & \vdots \\ T_{m,1} & T_{m,2} & \cdots & T_{m,m} \end{bmatrix} \tag{2}$$

The TAZ-based network of urban transit travel $G_{TAZ}$ was obtained using the TAZ mass centers as the nodes of the transit travel network, the line connections between TAZ mass centers were taken as the edges, and the volumes of transit between TAZs were taken as the edge weights:

$$G_{TAZ} = \{N_{TAZ}, E_{TAZ}, W_{TAZ}\} \tag{3}$$

where $N_{TAZ}$ is the set of nodes of $G_{TAZ}$ (i.e., TAZ mass centers), $E_{TAZ}$ is the set of network edges (i.e., the connecting lines between TAZ mass centers), and $W_{TAZ}$ is the edge weights (i.e., the transit travel volume between TAZs). When $W_{TAZ}$ is 0, the corresponding edge is not included in the network.

After building the urban transit travel network, we can describe the basic characteristics of the network with several basic statistical variables, including graph density, average clustering coefficient, and characteristic path length [28]. Of these, graph density reflects the edge connectivity of the network and the average clustering coefficient describes how closely the nodes in the network are related to each other, with higher values indicating greater edge connectivity and closer proximity of the nodes in the network. The characteristic path length indicates the average graph distance between all pairs of nodes in the network, and a smaller value indicates a more reachable and efficient network. These metrics can be used to characterize whether a network is a small-world network and whether it has an obvious community structure. A small-world network refers to a network in which most nodes can reach any other nodes through short communication paths, as judged by the clustering coefficient and the characteristic shortest path [29,30]. If the average clustering coefficient of the network is higher than the random network with the same number of nodes, and the characteristic path length is similar to the random network, the network is considered to have the small-world property, and the network has a community structure [29–31].

### 3.3. Transit Travel Community Detection

Based on graph theory, the community detection method in complex networks can be used to divide huge networks into several closely connected non-overlapping communities. The nodes in the community after division are closely connected, whereas those between communities are relatively loosely connected [32,33]. At the city level, community detection based on travel data helps reveal the spatial interaction patterns of urban travel activities and thus provides a decision-making basis for intelligent urban planning. In this study, the Louvain algorithm was used to detect transit travel communities [34], this algorithm is characterized by fast running speed, an intuitive detection process, and easy implementation. Additionally, no supervision is required for the obtained results and small communities will not be missed. The core concept of the Louvain algorithm is to continuously increase the modularity $Q$ of the whole network through repeated iterations to ensure a more reasonable division of communities. If the modularity value exceeds 0.3, this indicates that there is a clear community structure [35]. The definition of $Q$ is:

$$Q = \frac{1}{2m} \sum_{i,j} \left[ A_{ij} - \frac{k_i k_j}{2m} \right] \delta\left(c_i, c_j\right) \tag{4}$$

where $A_{i,j}$ represents the weight of the edge between network nodes $i$ and $j$, $k_i$ is the sum of edge weights connected to node $i$, and $m$ refers to the sum of weights of all edges in the network. In addition, $c_i$ represents the community in which node $i$ is located and $\delta(c_i, c_j)$ is used to determine whether nodes $i$ and $j$ are in the same community, returning a value of 1 if yes and 0 if no.

In the Louvain algorithm, there are two main detection phases for the structure of urban transit travel communities. In the first phase, each node in the network graph (the nodes in the initial network are TAZ mass centers) was regarded as a separate community. For each node $i$, the node was successively distributed to the communities in which each of its network neighbors was located, the change of network modularity $\Delta Q$ before and after distribution was calculated, and the node transfer between communities was evaluated to

allocate the node to the community corresponding to the largest $\Delta Q$ value. All nodes were processed sequentially in the first community detection phase. $\Delta Q$ is expressed as follows:

$$\Delta Q = \left[ \frac{\sum k_{in} + 2k_{i,in}}{2m} - \left( \frac{\sum k_{tot} + k_i}{2m} \right)^2 \right] - \left[ \frac{\sum k_{in}}{2m} - \left( \frac{\sum k_{tot}}{2m} \right)^2 - \left( \frac{k_i}{2m} \right)^2 \right] \tag{5}$$

where $\sum k_{in}$ represents the sum of the weights of the inner edges of the community, $\sum k_{tot}$ is the sum of the weights of all node edges in the community, and $\sum k_{i,in}$ denotes the sum of the edge weights from node $i$ to all the nodes in the community.

In the second phase, the graph was reconstructed. Each node in the new graph represented the mass center of each community detected in the first phase and the edge weights between communities were aggregated to form the new node edge weights. The process of the first phase was iterated repeatedly until the modularity of the whole graph no longer changed, thus forming a closely connected network community structure.

### 3.4. Transit Travel Community Evolution

This study analyzes the evolution characteristics of urban transit community from the perspectives of similarity and difference. For the similarity perspective, a similarity index is constructed to measure the evolution characteristics. For the difference perspective, visualization analysis is conducted based on Sankey and cumulative scale diagrams.

Among them, the similarity index quantifies the dynamic evolution characteristics of a transit travel community. For different years $u$ and $v$, the similarity of urban transit travel communities can be expressed as $S(u, v)$:

$$S(u, v) = \frac{\sum_i \delta \left( C_i^u, C_i^v \right)}{n} \tag{6}$$

where $n$ represents the total number of TAZs in the urban space units. For any $TAZ_i$, $C_i^u$ indicates the transit travel community category of $TAZ_i$ in year $u$. If the transit travel community category of $TAZ_i$ in year $u$ is the same as that in year $v$, then $\delta \left( C_i^u, C_i^v \right)$ is 1, otherwise $\delta \left( C_i^u, C_i^v \right)$ is 0.

## 4. Results

### 4.1. The Dynamic Network Structure of Urban Transit Travel

As mentioned above, the dynamic urban transit travel network in Shenzhen was constructed based on transit smart card data for weekdays and weekends from 2015 to 2017, with TAZs chosen as the analysis units. In addition, the basic network attributes were analyzed in terms of the graph density, average clustering coefficient, characteristic path length, and other indicators. Several random networks with the same number of nodes and edges have been constructed. Results show that the average clustering coefficient is approximately 0.25, and the average characteristic path length is approximately 1.5, whereas the average clustering coefficient values for weekdays and weekends from 2015 to 2017 were approximately 0.6, and the average shortest path length was approximately 1.7; these results indicate that Shenzhen's transit travel network has the small-world property and a clear community structure locally within the network (Table 1). The graph density and average clustering coefficient values of the transit travel network were the largest in 2017 and smallest in 2016. Within each year, the graph density and average clustering coefficient values of weekends were smaller than those of weekdays, indicating that transit travel in Shenzhen is more densely associated on weekdays than on weekends.

The overall spatial distribution of urban transit travel was further analyzed by mapping the average daily transit passenger flow on weekdays and weekends from 2015 to 2017. The average daily transit travel demand on weekdays and weekends in 2015, 2016, and 2017 are shown in Figure 3a and Figure 3b, respectively (Figure 3). As shown in Figure 3a, the transit passenger flow on weekdays and weekends was high in Nanshan District, Futian District, Luohu District, Longhua District, and the southern part of Longgang District, whereas values in other areas were relatively low, mainly as a consequence of the land utilization

and spatial structure of Shenzhen. Nanshan, Futian, and Luohu are centers of employment, entertainment, leisure, and culture, containing high-density living and entertainment facilities; thus, there is a high demand for transit travel in these areas. Furthermore, the transit infrastructure of these three districts is relatively complete and highly accessible, ensuring adequate public transport availability. In the Nanshan District Science and Technology Park, the transit travel demand on weekends was markedly different from that on weekdays, indicating that the work-related transit travel demand on weekdays changes significantly.

**Table 1.** Transit network attributes of Shenzhen.

| Year/Indicator | Weekday | | | | | Weekend | | | | |
|---|---|---|---|---|---|---|---|---|---|---|
| | Nodes | Edges | Graph Density | Average Clustering Coefficient | Characteristic Path Length | Nodes | Edges | Graph Density | Average Clustering Coefficient | Characteristic Path Length |
| 2015 | 491 | 67969 | 0.316 | 0.613 | 1.73 | 491 | 57066 | 0.267 | 0.591 | 1.798 |
| 2016 | 491 | 67271 | 0.31 | 0.609 | 1.733 | 491 | 42580 | 0.198 | 0.581 | 1.954 |
| 2017 | 491 | 71114 | 0.327 | 0.619 | 1.721 | 491 | 59742 | 0.275 | 0.606 | 1.801 |

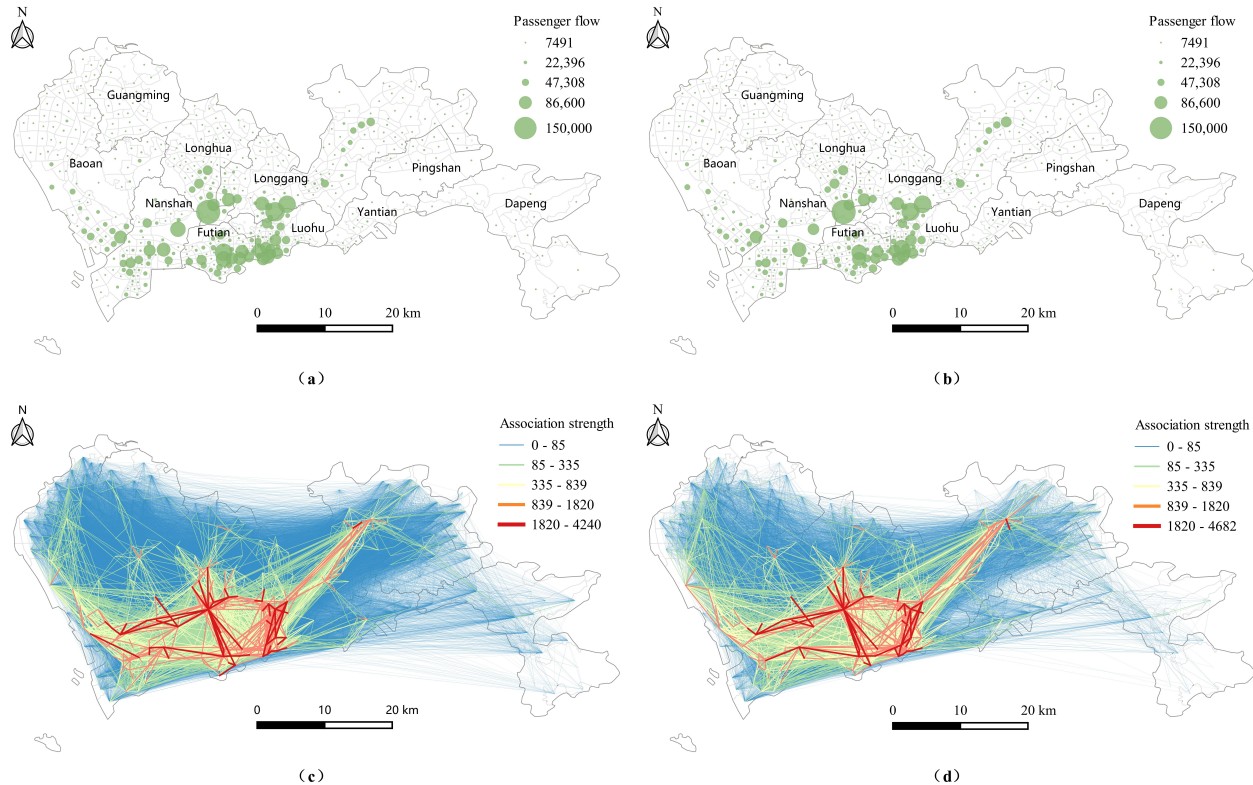

**Figure 3.** Spatial distribution of transit travel demand and daily transit passenger flow: (**a**) transit travel demand on weekdays; (**b**) transit travel demand on weekends; (**c**) daily transit passenger flow on weekdays; (**d**) daily transit passenger flow on weekends.

Using the Jenks breakpoint method, the transit travel association strength between TAZs was divided into five grades in ascending order. As shown in Figure 3c,d, a clear spatial agglomeration of transit travel is identified in the central city, i.e., a radial structure with Nanshan–Bao'an, Futian–Longhua, and Luohu–Longgang districts as its main axes was present in both weekday and weekend travel. In addition, there was also a prominent transit association axis between Bao'an–Longhua–Longgang and Nanshan–Futian–Luohu districts, indicating the presence of north–south and east–west urban transit corridors in Shenzhen. In comparison to weekdays, the association strength of urban transit activities on weekends was relatively small. For example, the flow of transit activity between Bao'an and Nanshan on weekdays is at grade 5 (red line in black circle in Figure 3c), whereas the flow between the regions on weekends drops to grade 4 (orange line in black circle in Figure 3d).

### 4.2. The Structure of Urban Transit Travel Communities

Community detection is an effective approach to identify the overall structural characteristics of urban space. By modifying the resolution parameters of community division, the modularity values were high when there were between 5 and 11 transit travel communities (Figure 4). Shenzhen has 10 district administrative units (excluding the Shenzhen–Shantou Cooperation Zone), given the consistency between the number of communities and administrative units, Shenzhen was divided into 10 communities for subsequent analysis in this study.

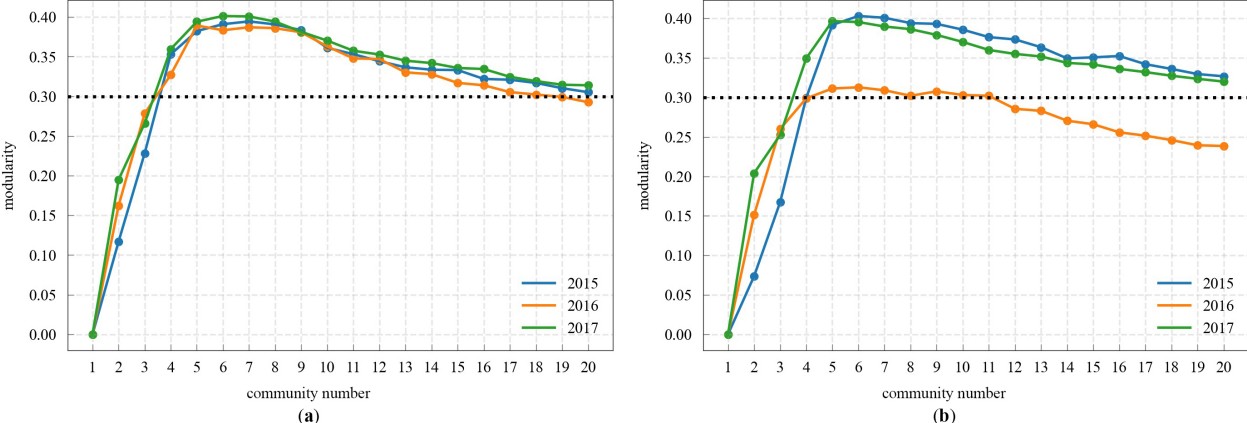

**Figure 4.** Plots illustrating the numbers of transit travel communities and their corresponding modularity values: (**a**) weekdays; (**b**) weekends.

The spatial distribution of transit travel communities on weekdays and weekends from 2015 to 2017 is shown in Figure 5. Overall, there were primarily spatial neighbor relations between TAZs in the same community, indicating that the city's transit travel corresponds largely with the geographical spatial effect. As illustrated by the structure of transit travel communities in three studied years, the community structure in different years varied greatly, and the structure of urban transit travel underwent dynamic change.

As shown in Figure 5a,b, most of the urban transit travel communities on weekdays were trans-district communities, whereas the western part of Bao'an District was an independent community. This area includes ports, airports, other large transport infrastructure elements, and manufacturing parks, serving as Shenzhen's main development area for its modern logistics industry and high-end manufacturing industries. The eastern part of Bao'an District, Guangming District, and the northern part of Nanshan District belonged to the same community, the other areas in Nanshan District comprised a new community in which the development of high-tech, educational research, and cultural industries dominates. Futian District was divided into two communities along its central part, where the northern part formed a trans-district community together with the TAZ of the Shenzhen North Railway Station in Longhua District. Luohu District was composed of two communities, in which the district's western part was closely linked to Futian District, whereas its eastern part constituted a community with Yantian District. The western part of Longgang District comprised an independent community, whereas the rest of Longgang District formed an ultra-large community with Pingshan District and Dapeng New District, in which industrial and ecological businesses are mainly present. There were some differences observed in the community structure on weekends and weekdays in 2015; specifically, the TAZs around Shenzhen North Railway Station were closely connected with the northern part of Futian District on weekdays, whereas they were more closely connected with its surrounding areas on weekends.

The southern part of Bao'an District and the southern part of Nanshan District belonged to the same community on weekdays in 2016, showing a significant difference relative to the transit travel community category in 2015 (Figure 5c), indicating that the transit association between Bao'an District and Nanshan District was significantly enhanced by the opening of Metro Line 11 in June 2016. In contrast, the southern part of Bao'an District and the southern

part of Nanshan District belonged to two different communities on weekends, implying that the newly opened metro line dominantly carries commuter passenger traffic (Figure 5d).

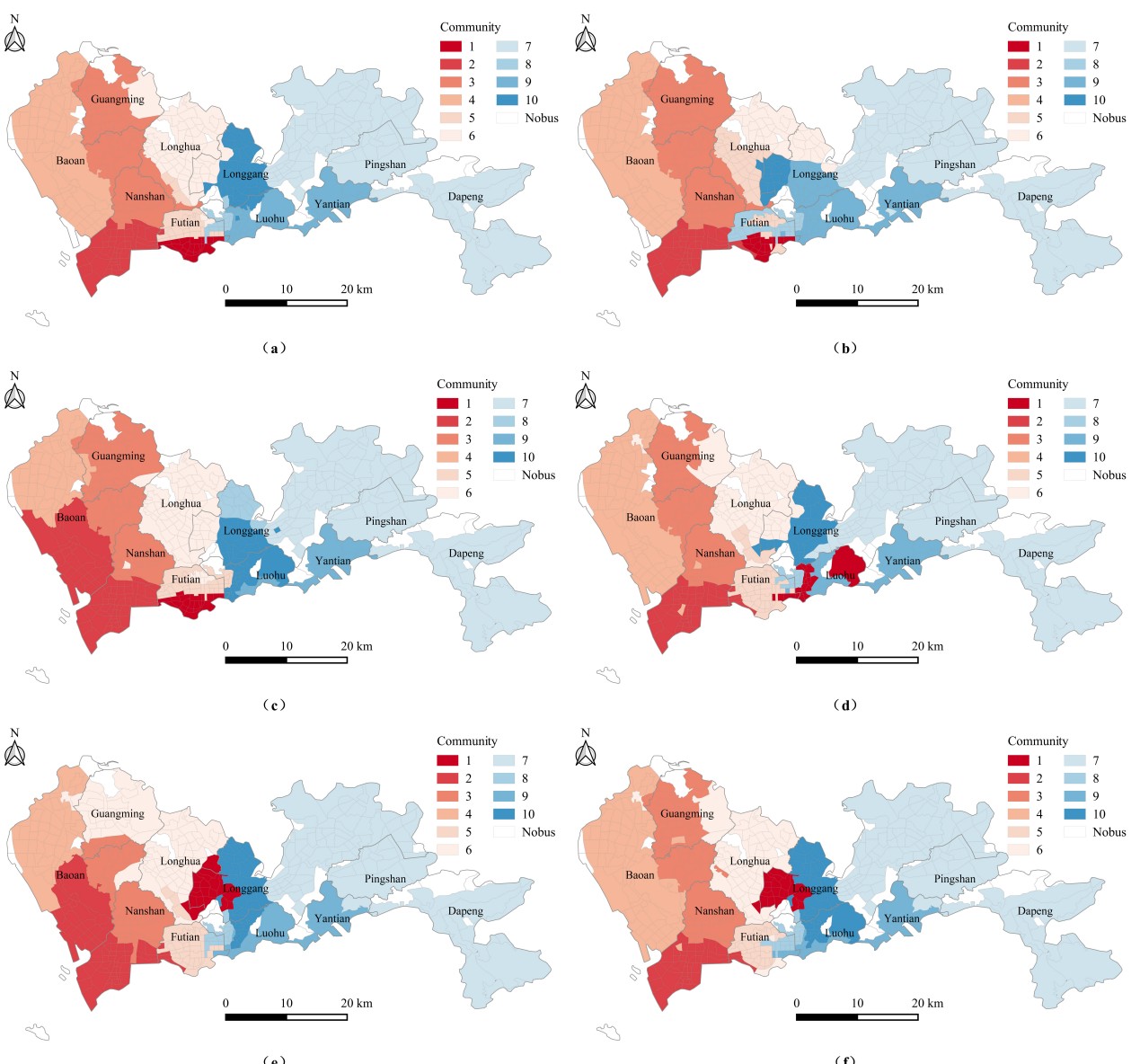

**Figure 5.** Transit travel communities in Shenzhen from 2015 to 2017: (**a**) weekdays in 2015; (**b**) weekends in 2015; (**c**) weekdays in 2016; (**d**) weekends in 2016; (**e**) weekdays in 2017; (**f**) weekends in 2017.

Relative to the communities in 2015 and 2016, the transit travel community category of Guangming District in 2017 changed significantly (Figure 5e). In 2015 and 2016, Guangming District, the eastern part of Bao'an District, and the northern part of Nanshan District belonged to the same community, whereas in 2017, Guangming District was more closely connected to Longhua District. In addition, in contrast to the previous two years, an independent community was formed by the northern and southern parts of Futian District in 2017, indicating that this district's internal association was enhanced by the opening of metro lines 7 and 9 in December 2016. The western part of Longgang District became more densely connected with Luohu District on weekends in 2017, as shown by the prominent longitudinal axis in Figure 5f, highlighting that the residents' travel association between Longgang District and Luohu District is strengthened on weekends.

### 4.3. The Dynamic Evolution Characteristics of Urban Transit Travel Communities

4.3.1. The Similarity of Urban Transit Travel Community Evolution

Similarity index values can be used to quantify the dynamic evolution characteristics of urban transit travel communities in different years, as calculated using Formula (6). As shown in Table 2, from 2015 to 2016, the similarity indexes of transit travel community structure on weekdays and weekends were 0.82 and 0.76, respectively. These values were both higher than the equivalent values of 0.77 and 0.74 for the 2015 to 2017 period, indicating that the community similarity in adjacent years within a certain time series is relatively higher and that urban transit travel communities evolve progressively. This indicates that the spatial structure of urban transit travel has a certain stability, but because the urban transit infrastructure and built environment are in constant change, the transit travel communities also have some variability across different years. This difference is closely related to the transit infrastructure and built environment, etc., and the difference between adjacent years will be smaller, so the urban transit travel community has a gradual evolution characteristic. Additionally, the community similarity index values for weekdays and weekends highlight that the similarity of transit travel communities on weekdays during the three years was lower, thus, the structural evolution of transit travel communities is more prominent on weekdays. This result shows that the urban transit travel structure is dynamically changing. This is crucial for planners' ability to carry out optimal configuration of the transit network and optimize the spatial pattern of urban transit travel.

**Table 2.** The similarity of transit travel communities in Shenzhen from 2015 to 2017.

| Years | Weekday | Weekend |
|---|---|---|
| 2015–2016 | 0.82 | 0.76 |
| 2016–2017 | 0.76 | 0.84 |
| 2015–2017 | 0.77 | 0.74 |

4.3.2. The Diversity in the Evolution of Urban Transit Travel Communities

The urban transit travel communities on weekdays and weekends are different from 2015 to 2017. Based on the stability of the community numbers, the characteristics of the urban transit travel communities were further analyzed in terms of both spatial evolution and community volume evolution.

(1)    The diversity in spatial evolution

To visualize the spatial evolution of urban transit travel communities, the community evolution trends for weekdays and weekends were mapped. In the time range of 2015–2017, these changes include: only changed in 2015–2016, only changed in 2016–2017, both 2015–2016 and 2016–2017 have changed, and no change from both 2015–2016 and 2016–2017 (Figure 6).

As shown in Figure 6a, on weekdays, the community changes significantly in the middle and western areas of Shenzhen, including Futian District, Luohu District, Guangming District, the south of Bao 'an District, and the south of Longhua District. These results indicate that changes in transit infrastructure over the three years exerted a significant impact on the spatial pattern of urban transit travel on weekdays. However, as shown in Figure 6b, the community category evolution trends on weekends differed from those on weekdays. On weekends, the community changes significantly in the middle area of Shenzhen, including Luohu District, the south of Longhua District, and the west of Longgang District. These findings confirm that transit infrastructure had less impact on the spatial pattern of residents' travel on weekends.

(2)    The diversity in the evolution of community structure

Sankey diagrams, also known as heat balance diagrams or energy flowcharts [36], consist of nodes and connecting lines, where the nodes indicate source points, intermediate nodes, and sink points of various information and resource flows, whereas the connecting lines represent the flow of information and resources between the nodes [37]. Thus, the

dynamic evolution characteristics of the studied communities on weekdays and weekends from 2015 to 2017 were presented on Sankey diagrams, where the node lengths symbolize the number of TAZs in the community categories and the connecting lines represent the trends linking the community categories (Figure 7). Overall, the community that had the largest volume in all three years was consistently Community 7, which includes Dapeng New District and Longgang District. Communities 1, 5, and 8 consistently had the smallest volumes; these areas are primarily situated in the central city of Shenzhen.

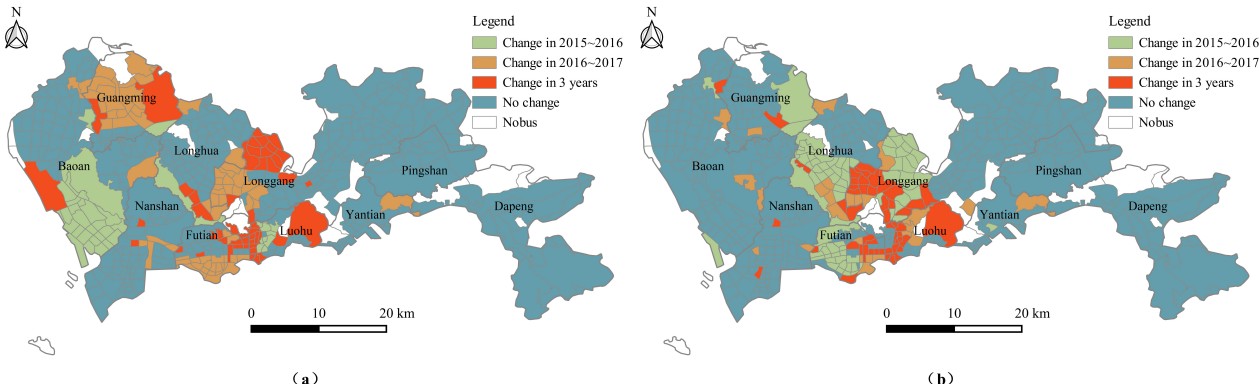

**Figure 6.** The evolution of transit travel communities in Shenzhen from 2015 to 2017: (**a**) weekdays; (**b**) weekends.

In terms of community change patterns, there were three typical transit travel community patterns on workdays: Community 4–Community 2–Community 2, Community 3–Community 3–Community 6, and Community 1–Community 1–Community 5. Among these, the first change pattern mainly involved the TAZs in the southern part of Bao'an District, whose community categories changed in 2016 and remained stable in 2017. This indicates that the opening of Metro Line 11 promoted connections between the southern part of Bao'an District and Nanshan District and their connections are now stable. The second and third change patterns mainly involved the TAZs of Guangming District and Futian District whose community categories remained stable in 2015 and 2016 but changed in 2017. On weekends, there were also three typical community change patterns: Community 5–Community 6–Community 6, Community 1–Community 5–Community 5, and Community 6–Community 10–Community 10. The corresponding community categories for all three change patterns changed in 2016 and remained stable in 2017.

Based on the transit travel volumes of each community category as a fraction of the city's total transit travel volume, the evolution of the relative transit travel volume was analyzed. The results are shown in Figure 8, in which the block colors represent the community categories and the block heights represent each community's proportion of transit travel volumes. As illustrated in Figure 8a, on weekdays, the relative transit travel volumes of each community category in the same year were unequal. Comparing travel volumes in different years indicates that the relative travel volumes of each community changed significantly in 2016. Specifically, the relative travel volumes of Communities 2 and 10 increased significantly and the volume of Community 9 decreased markedly in 2016. In contrast, in 2017 Communities 5 and 6 underwent the largest changes in their relative travel volumes. This analysis thus demonstrates that there are prominent differences in the relative transit travel volumes of each community category between the different years. As shown in Figure 8b, the relative travel volumes of each community within the same year were less unequal on weekends; the relative volumes for the same community in different years varied slightly, with near-equal relative volumes recorded for each community in 2017.

Furthermore, a comparison of Figure 7 and Figure 8 reveals that there were differences between the TAZ amounts of the communities and the transit travel volumes. For example, the TAZ volume of Community 7 was large in 2015; however, its relative transit travel

volume was less. This observation may be explained by the fact that Community 7 is mainly located within Longgang District and Dapeng New District—these two districts are large in area but have relatively low population density.

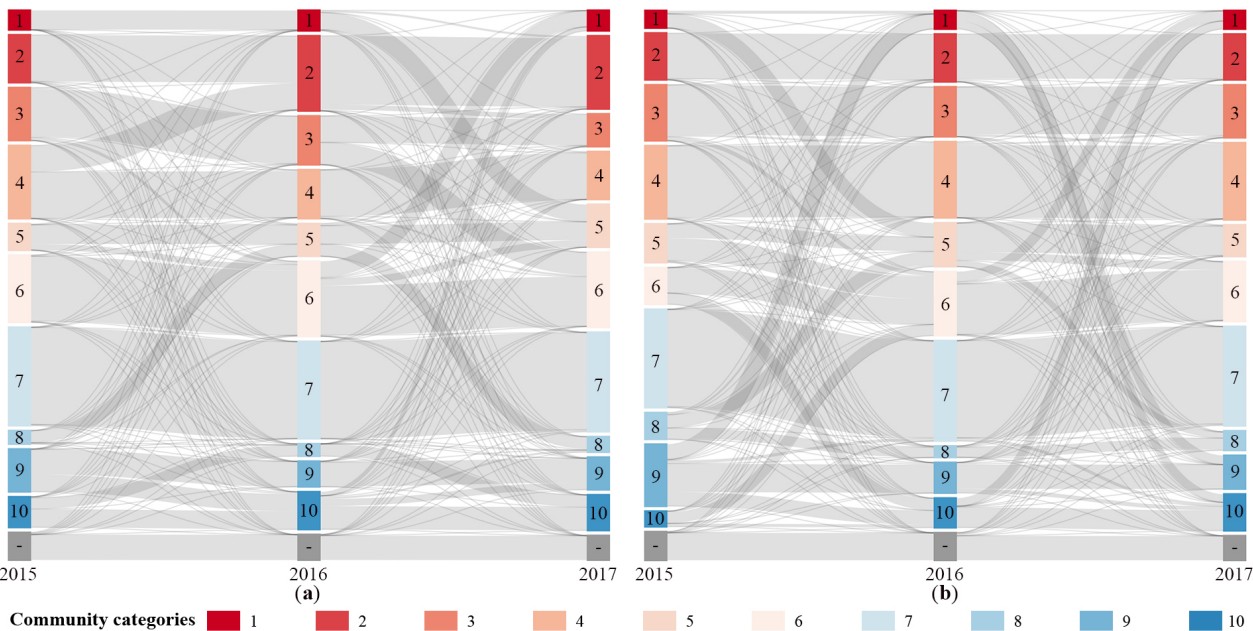

**Figure 7.** The Sankey diagrams of transit travel community changes in Shenzhen from 2015 to 2017: (**a**) weekdays; (**b**) weekends.

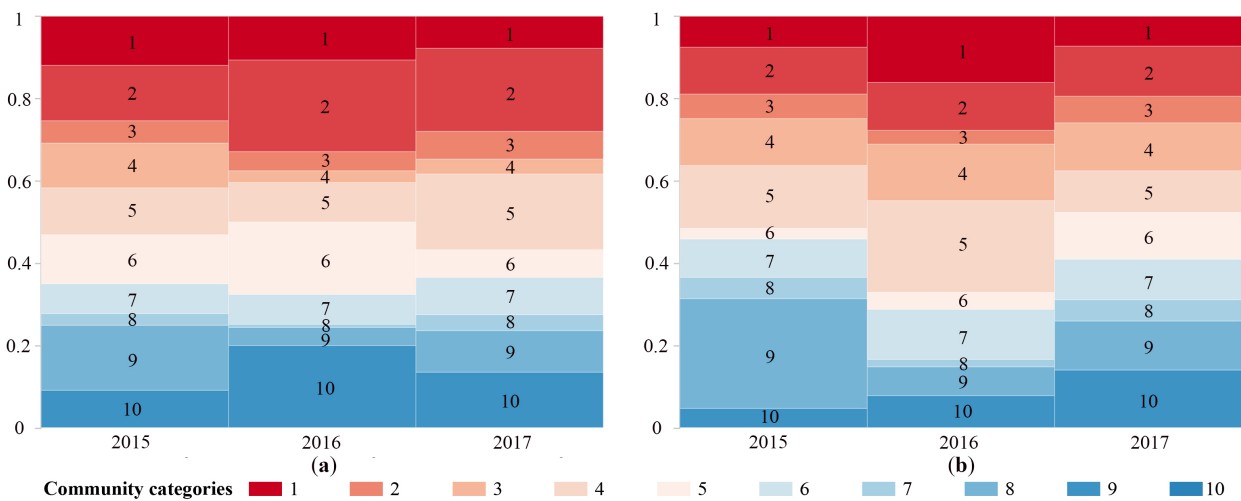

**Figure 8.** The relative travel volume distribution of transit travel communities in Shenzhen from 2015 to 2017: (**a**) weekdays; (**b**) weekends.

### 4.3.3. The Coupling Relationship between Transit Travel Communities and Administrative Units

The coupling relationship between the number of transit travel community categories and the number of urban administrative units was explored based on the premise that they were the same, as implied by the modularity analysis in Section 3.3. This relationship can be presented in terms of the link between communities and administrative districts as the matrix bubble diagram shown in Figure 9, in which the colors of the bubbles represent the community categories and the sizes of the bubbles represent the number of TAZs of the communities. As shown, the urban transit travel communities were linked to the boundaries of administrative districts, with each district largely associated with one main

community. Pingshan District, Yantian District, and Dapeng New District were each linked with only one community, reflecting the consistency of transit travel within the district. The other administrative districts were mostly linked to two or three communities. On weekdays in 2015, Bao'an District consisted of Communities 4 and 3, with Community 4 as its main community. Longgang District consisted of Communities 6, 7, and 10, with Community 7 as its main community. Additionally, the number of communities contained in each administrative district varied between years, for example, Bao'an District contained two communities (Communities 3 and 4) on weekdays in 2015 and three communities (Communities 2, 3, and 4) on weekdays in 2016 and 2017.

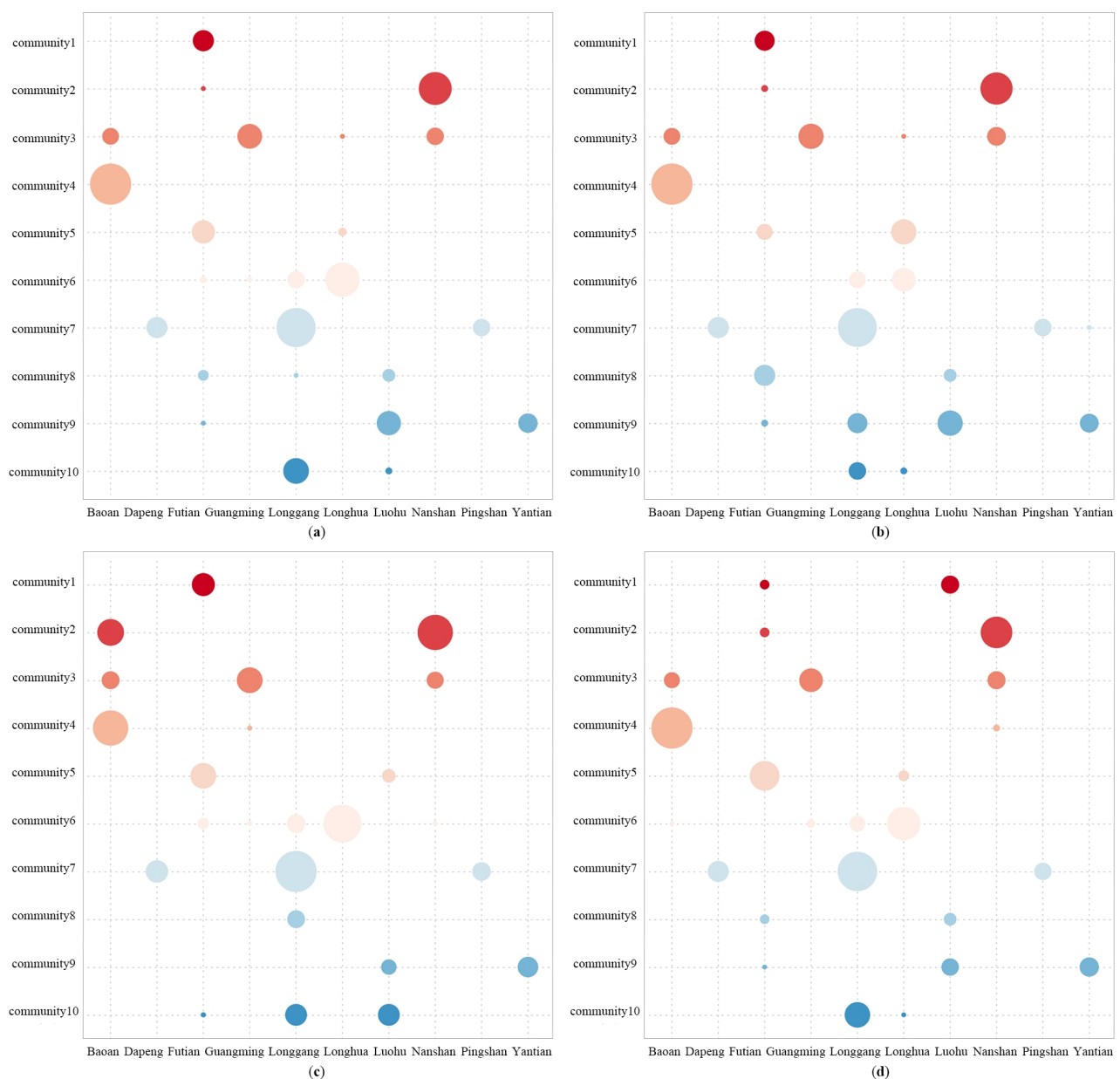

**Figure 9.** *Cont.*

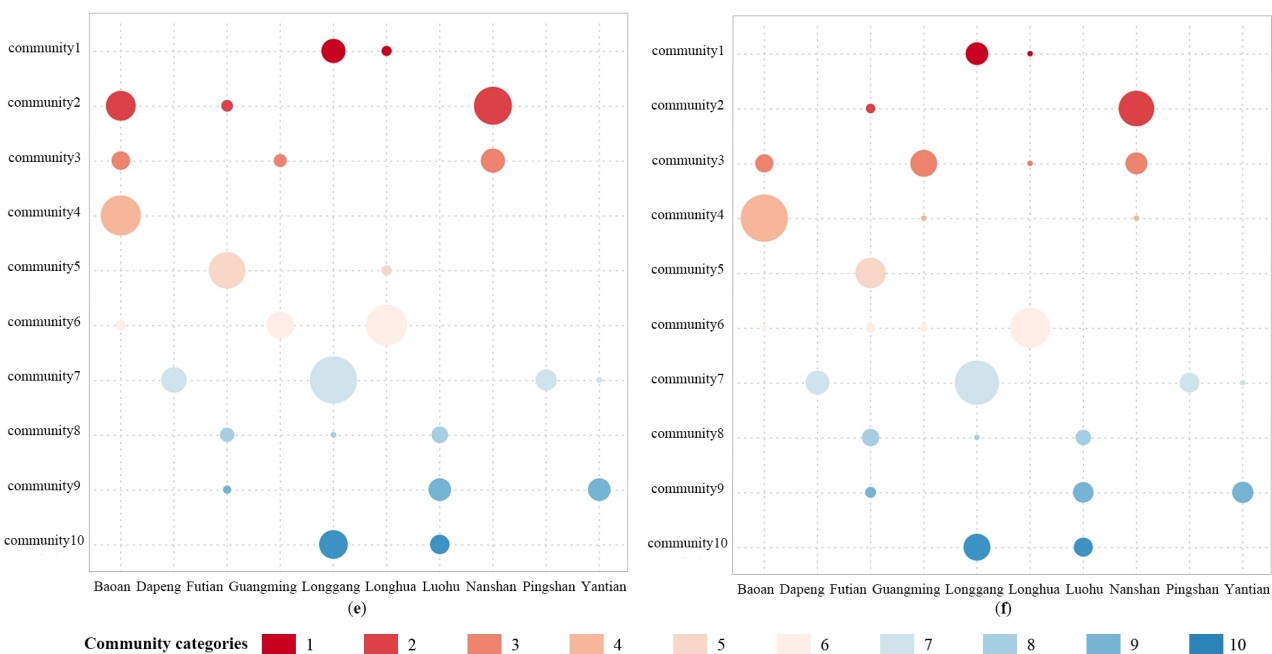

**Figure 9.** The comparison of transit travel communities and administrative districts in Shenzhen from 2015 to 2017: (**a**) weekdays in 2015; (**b**) weekends in 2015; (**c**) weekdays in 2016; (**d**) weekends in 2016; (**e**) weekdays in 2017; (**f**) weekends in 2017.

## 5. Discussion

In the era of spatiotemporal big data, the composition of spatial communities is expanding from static division in traditional planning to dynamic detection of actual activities so as to promote the dynamic cognition of urban spatial structure. Based on smart card data, this study explores the structure and evolution characteristics of urban transit travel communities in Shenzhen. Smart card data have clear geographical identifiers and time tags which reflect the daily travel characteristics of transit travelers and have obvious advantages for describing the collective network structure. In contrast to the urban planning department, the community structure based on transit travel provides an immediate reflection of the urban spatial structure and how it echoes or differs from the urban administrative district. Overall, the results of this study can provide guidance for urban land use planning and urban public transport planning. For example, after understanding the community structure of residents' transit travel, the transportation department can promote the connection between various regions of the city and promote the integration of urban transportation based on administrative means.

Some previous studies have applied different data sources to analyze the spatial structure of travel communities in Shenzhen and their coupling relationship with the administrative units. A study based on the jobs–housing relationship data divided Shenzhen into seven communities, with the jobs–housing communities interpreted to essentially follow the boundaries of urban administrative units [38]. Another study based on mobile signaling data divided Shenzhen into 32 communities, with Futian District as a complete community [21]. Based on transit travel data, in this study, the northern part of Futian District and the Shenzhen North Railway Station in Longhua District were distributed into one community on weekdays in the context of 10 communities in total, reflecting the spatial interaction characteristics of transit in urban commuting travel and the importance of analyzing the space structure of communities from a transit perspective.

Compared with previous studies that detected communities at a specific time, this study analyzes the evolution of communities in different years from the perspective of time series and then finds the evolution characteristics of communities. In the three years of our study, the community category changed significantly in 2016 because three subway

lines were newly opened in 2016. In addition, we found that Guangming District, the eastern part of Bao'an District, and the northern part of Nanshan District were closely connected, forming a clear longitudinal axis. Further improvement of the north–south transit infrastructure between these regions and the southern part of Nanshan District may also strengthen the association between Guangming, Bao'an, and Nanshan District, promoting the formation of a longitudinal axis running through the entirety of Shenzhen. The transit associations within Shenzhen may also be further enhanced by the opening of Shenzhen Metro Line 13, which is currently under construction in Nanshan District, Bao'an District, and Guangming District.

There are some limitations in this work. Due to the limitation of data, this study did not analyze transit travel community structure in longer period of years or for different seasons. Future work may disclose these temporal changes. Furthermore, it is worthwhile to explore, compare and integrate communities from other perspective, such as mobile phone data and travel navigation data. In addition, by incorporating higher granular population and economic data, the formation mechanisms of community may be better disclosed.

## 6. Conclusions

Based on Shenzhen's transit smart card data from 2015 to 2017, this study constructs an transit travel network, identifies transit travel communities using community detection methods, explores the evolutionary characteristics of transit travel communities from 2015 to 2017, and compares transit travel communities with urban administrative districts. The findings of the study highlight four key aspects:

(1) There is an obvious community structure of transit travel network in Shenzhen during 2015–2017.
(2) During 2015–2017, the community evolution is significant in the western and central parts of Shenzhen on weekdays, and in the central part of Shenzhen on weekends.
(3) The number of TAZs contained in different categories of communities and the amount of transit trips varied significantly across years, showing non-equilibrium characteristics.
(4) The transit travel communities and the administrative units are spatially coupled, with most administrative districts containing a main community.

This study makes the following contributions: (1) The study of spatial communities from the perspective of public transit travel is conducive to enriching the spectrum of urban spatial structure research supported by big data. (2) Transit travel communities may highlight the matching effect between travel demand and transit supply, which is beneficial to the synergistic development of urban land use and public transit, especially rail transit. (3) This study provides a detailed understanding of the dynamic urban spatial structure in both time and space, which will contribute to intelligent decision-making in urban spatial planning and transportation planning, ultimately promoting sustainable urban development.

**Author Contributions:** Conceptualization, J.Y. and Z.H.; methodology, J.Y., Z.H. and T.Z.; validation, J.Y. and T.Z.; writing—original draft, J.Y.; writing—review and editing, J.Y., Z.H., T.Z., Y.Z. and F.C. All authors have read and agreed to the published version of the manuscript.

**Funding:** This research was funded by the National Natural Science Foundation of China (NSFC) [Number: 42071357, 41901389].

**Institutional Review Board Statement:** Not applicable.

**Informed Consent Statement:** Not applicable.

**Data Availability Statement:** Not applicable.

**Conflicts of Interest:** The authors declare no conflict of interest.

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
