# Peer review of "Transit Travel Community Detection and Evolutionary Analysis: A Case Study of Shenzhen"

_sustainability, doi:10.3390/su15075900_

Round 1

Reviewer 1 Report

This study builds urban transit travel network based on smart cards data and TAZs, and applies complex network community detection methods to examine the dynamic evolution of urban transit travel communities in Shenzhen from 2015 to 2017. But this work can be improved as follows:

(1)  Introduction

Urban public transit system is a complex network. However, the study does not mention it until the “Results” section. I think authors should complement this point.

(2)  P 4

In Formula 2, the second row and second column of the matrix should be T2,2, not T2,1.

(3)  P 6, lines 171 and 172

() should be ).

(4)  P 6, line 183

The small-world network is presented by Watts and Storgaze (1998). I think authors should cite this reference in the paper.

(5)  P6, line 187

“The average shortest path is similar to that of the random network”. I think this needs to be checked since a small-world network means a higher clustering coefficient and average shortest path length.

(6)  P 6, lines 194-5

Why these results indicate that Shenzhen’s transit travel network has a small-world property? I think authors should add some references to make readers much clearer.

(7)  P 6, line 205, P 7, Figure 2

Figure 2 depicts the spatial distributions of the average daily passenger flow intensity of transit travel on weekdays and weekends from 2015 to 2017. Therefore, Figure 2 should include average daily transit passenger flow in Shenzhen. Otherwise, there would be misleading due to the expression “from 2015 to 2017”.

(8)  P 7, lines 224-225

“In comparison to weekdays, the association strength of urban transit activities on weekends was relatively small. For example, the association strength of transit activities between Bao’an and Nanshan on weekdays was strong (fifth grade in the Jenks breakpoint method), while the association strength between these areas on weekends was reduced to the fourth grade.”

This result is difficult to be observed in Figure 2c and 2d.

(9)  P 10, Lines 304-305

Here, TAZs are used to represent urban transit travel communities, not the transit travel communities identified by community detection methods. I think authors should explain it and avoid misunderstandings to the readers. In Figure 6, the number of transit travel communities is 10.

(10) P 10, Figure 5

Why there four different kinds of the spatial evolution of transit travel communities in Shenzhen from 2015 to 2017? What does 2015-2017 evolution mean?

Since the interpretation of the results is mainly based on administrative districts, the name of these districts might be added in Figure 5.

Please see the attachment for more details.

Reviewer 2 Report

The paper discusses an interesting and important topic. Urban public transit is a essential factor for promoting urban sustainable development. This research makes use of community diction algorithm to derive transit community and explore the characteristics of its evolution, which provides a different perspective from what we have learned using other types of mobile data.

I have a few pieces of feedback for the authors:

1.     Please clarify the meaning of “community” used in this study, e.g., travel community, urban community, which confuses the reader. Be consistent in the way you describe each type of community throughout the entire manuscript, and avoid using the single word “community”.

Also, the term “urban administrative units” used in this study is not clearly defined/explained.

2.     Line 3: what is the definition of “fairness”? does this mean “equality” or “equity”?

3.     Line 21 “urban communities are the regions….”, can “region” be used to describe “urban communities”?

4.     “Abstract” section: the information on the aim and novelty (the gap filled by this study) is not clarified. Please rephrase the abstract, by stating background, what has been done, the research gap, novelty of this study, methodology, findings, and contributions.

5.     The “Introduction” section needs to pay attention to the logical structure and clarifying the focus of each paragraph. The last paragraph should clarify the aim and novelty at the beginning, and then elaborate on the research contents.

6.     Line 113-120: authors mentioned that the smart card data are bus and metro card payment data, and the time information is card payment time. How did authors infer the check-in and check-out time for each trip using card payment time? Does this “card payment time” mean “check-in” or “check-out time”?

7.     Line 122: how the “abnormal value” was defined?

8.     Line 129-131: what is the definition of “regular users” and “non-regular users”?

9.     Line 139: what is the definition of “single-trip travel chains”?

10.   Line 175-187 and Line 230-239: the contents of these paragraphs may appear earlier in the manuscript, as they relate to methodology.

11.   Line 415-520: what can we learn from “the similarity of transit travel communities in adjacent years is high during the three studied years, with progressive evolution observed in the communities”? does this finding has any implications for planning and management?

Reviewer 3 Report

I appreciate the opportunity to review the article titled Transit Travel Community Detection and Its Evolution: A Case Study of Shenzhen.  I also appreciate author’s ability to choose an insightful study, particularly one that focuses on Transit Travel in an urban settings. After careful reading, I'd say the article has merit to be published in the journal but need to revise some content to make it clear to global reader. Having said that, I have recommendations that I believe the authors will find useful and that, if implemented, would make this research suitable for publication in the journal.

 #Title

Title of the article is somewhat unclear for a wider audience, author could consider revise the title of the article

 # Abstract

 The study's implication is unclear. At the end of the abstract, kindly include a statement outlining the study's implications. Methods and possible outcomes are not expressed academically.

 # Introduction

 Introduction is written poorly without proper background and discussion of previous related studies and most importantly the research gap which helped to formulate this particular research. I do not see the idea of the article has been established properly. Without scientific referencing and detailed analysis of those approaches, the conclusion and argument of the article do not sound appropriate. I recommend the author to look at the latest scientific publication before jumping to any conclusion, as a lot might have been changed in the decade-long period. In other way I would say in the Introduction, it is necessary to reconstruct the storyline of this manuscript. The introduction is the opening part of the scientific story to attract an audience and suggest the direction of your research. For this reason, you need to identify the problem that drives the research and introduce the key characters. If then, using the key characters, it is required to intertwine the scientific story concisely, systematically, and logically. Please rearrange the keywords so that the background of the study sounds appropriate.

 # Study Area

Provide a rationale why this area is selected for the study, too much of information need to shorten. Please change the study area map and make it in global context with proper index map otherwise readers would not be able to identify the study area.

 # Materials and method

 Entire methodology section is poor and vague which make the research unclear in terms of methodological design. It is hard to understand the direction of this research. It is necessary to reorganize the hierarchy of Materials and Methods such, overview of the method, Data Collection – source of the data must be included and need a rationale why authors used only three years data from (2015-2017), and Data Analysis- need to mention relevant geospatial process being adopted for this study – what sort of analysis being used for spatial mapping,. Please include a brief flowchart to help readers who are unfamiliar with this topic comprehend the study methodology. I do not see any flowchart for this study, please provide a brief but robust flowchart so that reader can be benefited from the study. Please make a lucrative chart using relevant graphics and symbol.

 # Result and discussion

 Result section seems to be OK but I found plagiarized sentences in between 175 to 199. Figure 2, 4 and 5 can be enlarged so that each map elements could be perfectly visible. As per as the discussion section is concern   This article does not have a proper scientific discussion section from where the reader can get comparative discussion on data and techniques relative to transit travel community.  Please elaborate limitation, future direction and policy implication of this study

 # Conclusion

 Should be revised, too much of generic sort of discussion here, I cannot see the brief of results of this study here. Please revise conclusion accordingly, so that reader could recall about the research again when they would be at the end of the article. Please remove limitation stuffs to discussion section.

Round 2

Reviewer 1 Report

The paper has been much improved. However, there are still several grammatical mistakes. For example, "Among existing researches" should be "Among existing research" in line 2. 

Please see the attachment for more details to be checked and revised.

Reviewer 3 Report

I am satisfy with the edits

Author Response

Thank you for your approval of our revision.